# Application of Molecular Hydrogen as an Antioxidant in Responses to Ventilatory and Ergogenic Adjustments during Incremental Exercise in Humans

**DOI:** 10.3390/nu13020459

**Published:** 2021-01-30

**Authors:** Ahad Abdulkarim D. Alharbi, Naoyuki Ebine, Satoshi Nakae, Tatsuya Hojo, Yoshiyuki Fukuoka

**Affiliations:** 1Graduate School of Health and Sports Science, Doshisha University, Kyoto 610-0394, Japan; cyhf0001@mail4.doshisha.ac.jp (A.A.D.A.); nebine@mail.doshisha.ac.jp (N.E.); thojo@mail.doshisha.ac.jp (T.H.); 2Division of Bioengineering, Graduate School of Engineering Science, Osaka University, Osaka 560-8531, Japan; snakae@bpe.es.osaka-u.ac.jp

**Keywords:** hydrogen supplement, acid status, muscle deoxygenation, ventilation, incremental exercise

## Abstract

We investigated effects of molecular hydrogen (H_2_) supplementation on acid-base status, pulmonary gas exchange responses, and local muscle oxygenation during incremental exercise. Eighteen healthy, trained subjects in a randomized, double-blind, crossover design received H_2_-rich calcium powder (HCP) (1500 mg/day, containing 2.544 µg/day of H_2_) or H_2_-depleted placebo (1500 mg/day) for three consecutive days. They performed cycling incremental exercise starting at 20-watt work rate, increasing by 20 watts/2 min until exhaustion. Breath-by-breath pulmonary ventilation
($\dot{V}$_E_) and CO_2_ output
($\dot{V}$CO_2_) were measured and muscle deoxygenation (deoxy[Hb + Mb]) was determined via time-resolved near-infrared spectroscopy in the *vastus lateralis* (VL) and *rectus femoris* (RF). Blood gases’ pH, lactate, and bicarbonate (HCO_3_^−^) concentrations were measured at rest and 120-, 200-, and 240-watt work rates. At rest, the HCP group had significantly lower
$\dot{V}$_E_,
$\dot{V}$CO_2_, and higher HCO_3_^−^, partial pressures of CO_2_ (PCO_2_) versus placebo. During exercise, a significant pH decrease and greater HCO_3_^−^ continued until 240-watt workload in HCP. The
$\dot{V}$_E_ was significantly lower in HCP versus placebo, but HCP did not affect the gas exchange status of
$\dot{V}$CO_2_ or oxygen uptake ($\dot{V}$O_2_). HCP increased absolute values of deoxy[Hb + Mb] at the RF but not VL. Thus, HCP-induced hypoventilation would lead to lower pH and secondarily impaired balance between O_2_ delivery and utilization in the local RF during exercise, suggesting that HCP supplementation, which increases the at-rest antioxidant potential, affects the lower ventilation and pH status during incremental exercise. HPC induced a significantly lower O_2_ delivery/utilization ratio in the RF but not the VL, which may be because these regions possess inherently different vascular/metabolic control properties, perhaps related to fiber-type composition.

## 1. Introduction

Molecular hydrogen (H_2_) is a colorless, tasteless, odorless, and minimal molecule with high flammability [1]. Most mammals, including humans, do not synthesize hydrogenase, which is a catalyst for the activation of H_2_ [2]. In sports science, there is limited research regarding the antioxidant effect of H_2_ on exercise-induced oxidative stress. Unlike conventional antioxidants, H_2_ is a gas molecule and as such it is believed to have several advantages for application in sports science [3,4,5]. H_2_ is the smallest molecule and thus can penetrate the cellular membrane and rapidly diffuse into organelles (e.g., mitochondria). H_2_ is thought to have no effect on physiologically reactive species (e.g., H_2_O_2_), as it can selectively reduce •OH and ONOO^−^ [6]. In addition, H_2_ can be supplied to the body through multiple routes of administration, such as the oral intake of H_2_ water, H_2_ bathing, an intravenous infusion of H_2_ saline, and the inhalation of H_2_ gas.

In addition to these advantages, H_2_ can be used with minimal side effects as it is excreted by exhaling. If H_2_ is absorbed well in the intestinal tract, it will lead to an increase in bicarbonate ion (HCO_3_^−^) as well as a reduction in reactive oxygen species (ROS). Aoki et al. (2012) initially reported that in trained young men, an oral intake of H_2_ water suppressed the elevation of blood lactate concentrations and reduced peak torque during exercise [7]. Thus, H_2_ water leads to a strong buffering capacity to glucose metabolism and/or exercise performance. We, thus, hypothesized that the intake of H_2_ would promote buffering lactic acid and lead to higher pH, which would induce delayed metabolic acidosis and extend exercise performance. The prior studies mostly used H_2_ in gas or infused water forms, but we developed a new chemical engineering technique for hydro-calcium powder (HCP) that uses extracted calcium to absorb highly concentrated hydrogen molecules (International Patent No. 4,472,022).

In investigations of the relationship between muscle O_2_ delivery
($\dot{Q}$O_2_) and oxygen uptake (
$\dot{V}$O_2_), time-resolved near-infrared spectroscopy (TR-NIRS) was able to resolve the absolute values of [heme] chromophores [8] and provided improved mechanistic insights into the effects of HCP on muscle
$\dot{Q}$O_2_-to-
$\dot{V}$O_2_ relationships. It is also presently unclear whether dietary HCP supplementation might alter the muscle spatial heterogeneity in muscle deoxygenation (deoxy[Hb + Mb]) kinetics (and, by extension,
$\dot{Q}$O_2_-to-
$\dot{V}$O_2_ matching) that was observed during cycling exercise [8,9,10,11,12,13,14,15]. HCP may increase HCO_3_^−^ and delay metabolic acidosis. As described above, since HCP causes opposite effects at the same time, the extent of change that HCP causes in the muscle spatial heterogeneity in deoxy[Hb + Mb] kinetics at different muscle sites is not yet known.

The aforementioned advantages of H_2_ use are expected to be a potential scavenger of superoxide anions, as it was reported that the augmented hypoxic carotid body sensory response was abolished in a group of rats pretreated with a superoxide dismutase mimetic [16,17]. Wilkerson et al. also demonstrated that the intravenous administration of a superoxide anion scavenger to anesthetized, vagotomized, and ventilated rats before their exposure to acute hypoxia abolished the long-term facilitation (LTF) of phrenic nervous activity [18]. Given these findings in animals, one of the possibilities is that ROS are downregulated by the administration of molecular hydrogen, resulting in hypoventilation and blood hypercapnia. We, therefore, hypothesized that the administration of a strong antioxidant of molecular hydrogen would reduce the magnitude of ventilatory and deoxy[Hb + Mb] responses before and during exhaustive exercise in humans.

To test these contradictory hypotheses, we conducted the present study to determine whether an HCP supplement causes different responses to muscle deoxy[Hb + Mb] and the responses of pulmonary ventilatory and blood gas values at both rest and incremental exercise compared to placebo.

## 2. Subjects and Methods

### 2.1. Subjects

Eighteen healthy males with no cardiac, ventilatory, gas exchange, or musculoskeletal disorders participated. Their mean ± SD values were age, 21 ± 1 years old; height, 174.0 ± 4.4 cm; and body weight, 66.6 ± 6.3 kg. Written, informed consent was obtained from all subjects after a detailed explanation of all procedures, the purpose of the study, the possible risks, and the benefits of participation. Our study conformed to the Declaration of Helsinki, and the ethical committee of Doshisha University approved all study procedures (No. 18012).

### 2.2. Supplements

Each subject was examined twice in a crossover, double-blind manner and given four capsules/day for three days of either H_2_-rich calcium powder (HCP) supplements (ENAGEGATE, Tokyo) or calcium powder as a placebo (ENAGEGATE). The capsules for both HCP and placebo were identical, with a clear outer shell and white powder fill inside. Each capsule weighted approx. 633 mg, with 375 mg of either HCP or only calcium powder for the placebo. The amount of hydrogen in the HCP was 0.636 µg/capsule. The subjects were instructed to consume the supplements in one dose of four capsules at 9:00 p.m. (±1 h) for the three consecutive days prior to the experiment day, amounting to the total H_2_ amount of 2.544 µg/day.

### 2.3. Experimental Overview

The subjects were familiarized with all of the measurement techniques, including the cycling exercise with a face mask. A familiarization test was conducted 1–2 weeks prior to the start of the experiments, and the experiments were separated by a six- to seven-day washout/rest period.

On the day prior to testing, the consumption of alcohol and caffeine was prohibited, and when the exercise or training was carried out, the intensity, the timing, and duration were matched for both experiments. The subjects were also instructed to record and replicate their dietary intake for dinner the previous night and for breakfast before the testing. The subjects consumed breakfast at home ≥6 h prior to the start of the experiment (and were instructed to replicate it from their dietary intake report). At 3 h before the experiment, the subjects ate a small meal consisting of the Calorie Mate (four blocks, Otsuka Pharmaceutical, Tokyo) and one bottle of caffeine-free barely tea (Healthy Mineral Barley Tea, 600 mL, ITO EN, Tokyo) that was standardized for all subjects, in order to avoid hunger and minimize fluctuations in significant blood metabolic parameters (specifically blood glucose) among/within subjects.

All studies were conducted in a custom-made environmental chamber (LP-2.5PH-SS, NKsystem, Osaka, Japan) maintained at a temperature of 25 °C with 50% relative humidity with minimal external stimuli.

The subjects performed an incremental exercise test using a cycle ergometer (75XL-III; Konami, Tokyo). The exercise started with 2 min at the workload of 20 W, after which the workload was increased at 20 W/2 min until the subject’s exhaustion or 300 W was reached, and the subjects were instructed to maintain their pedal frequency at 60 rpm throughout the exercise. When given criteria were met (e.g., a plateau or a drop in
$\dot{V}$O_2_, a heart rate (HR) > 95% of the age-predicted maximum [19], or a respiratory exchange ratio > 1.1), the highest average value of 1-min
$\dot{V}$O_2_ was regarded as the individual’s peak oxygen uptake [20].

### 2.4. Analyses of Blood Metabolites

A peripheral venous catheter was placed in the subject’s forearm to allow free movement of his elbow and hands during the experiment, and a 100-µL blood sample was collected at four time points: pre-exercise and at 120, 200, and 240 W during the incremental exercise. The blood samples were analyzed for blood gas, electrolytes, and the metabolic profile with a portable blood analysis system (epoc^®^, Siemens Healthcare, Tokyo) for the determination of blood gases, acid status of pH, bicarbonate (HCO_3_^−^), and blood properties of hemoglobin (Hgb) and hematocrit (Hct). HCO_3_^−^ was calculated from the partial pressures of CO_2_ (PCO_2_) and pH values according to the Henderson–Hasselbalch equation, and the base excess (BEecf) was calculated according to the following equation [21]:BE = (1 − 0.014 × [Hb]) × ([HCO_3_^−^] − 24.8 + (1.43 × [Hgb] +7.7) × (pH − 7.4))

The metabolic status of lactate (Lac), glucose (Glu), and creatinine (Crea), the electrolytes of the serum sodium (Na^+^), potassium (K^+^), chloride (Cl^−^) concentrations, • and the Aniongap (AGap) and Aniongap potassium (AGapK) were calculated by electrolyte parameters.

### 2.5. Measurements

Each subject’s pulmonary gas exchange was measured breath by breath throughout all tests as described [13,22,23]. The breath-by-breath gas exchange system (AE-310s; Minato Medical Sciences, Osaka) was calibrated according to the manufacturer’s recommendation before each exercise test. The subject breathed through a lower-resistance mouthpiece connected to a hot wire flowmeter for the measurement of inspiratory and expiratory flow and volume. Inspired and expired gases were continuously sampled from the subject’s mouth, and the O_2_ and CO_2_ fractional concentration were measured by fast-responding paramagnetic and infrared analyzers, respectively. The gas volume and concentration signals were time-aligned to account for the time lag between the signals for the calculation of the gas exchange parameters on a breath-by-breath basis. Alveolar gas exchange variables were calculated according to the algorithms of Beaver et al. [24]. The breath-by-breath
$\dot{V}$_E_ (BTPS),
$\dot{V}$O_2_ (STPD),
$\dot{V}$CO_2_ (STPD), gas exchange ratio (R), and end-tidal CO_2_ pressure (P_ET_CO_2_) were determined.

The electrocardiogram (ECG), taken from a *V5* lead, was monitored continuously on a wireless ECG monitor (DS-2150; Fukuda Denshi, Tokyo), and subject’s HR was measured by beat-by-beat counting of the R-spike of the ECG taken simultaneously with the other measurements.

The absolute values of oxygenated (oxy[Hb + Mb]) and deoxygenated (deoxy[Hb + Mb]) and the total hemoglobin and myoglobin concentration (total[Hb + Mb]) were sampled from the *vastus lateralis* (VL) and *rectus femoris* (RF) muscles of the subject’s dominant leg by a TR-NIRS system (C12707, Hamamatsu Photonics, Hamamatsu, Japan). This system measures the distribution of in vivo optical path lengths, thereby enabling the determination of the absolute [Hb + Mb] concentration (μmol∙L^−1^). The deoxygenation measured by the TR-NIRS was demonstrated to correlate significantly with the oxyhemoglobin saturation in both the blood and a purified-hemoglobin phantom solution [10,25,26].

The optodes were housed in black rubber holders that helped to minimize extraneous movement, thus ensuring that the position of the optodes was fixed and invariant. The distal optodes were placed on the lower third of the VL and the RF muscles parallel to the major axis of the thigh. The location of the distal optodes on the VL muscle was chosen to represent the single-site TR-NIRS measurement conducted in previous studies [27,28,29]. The proximal optode pairs on the VL and the RF muscles were located ~10–15 cm from the distal optode pairs. The interoptode spacing between the emitter and the receiver was 3 cm. The depth of the measured area was assumed to be approximately one-half of the distance between the emitter and the receiver, ~1.5 cm.

The skin under the probes was carefully shaved. Pen marks were made on the skin to indicate the margins of the rubber holder to check for any downward sliding of the probe during cycling and for accurate probe repositioning on subsequent days. No sliding was observed in any subject at the end of each protocol. The principles of operation and the algorithms used by the equipment are described in detail elsewhere [30,31]. Calibration of both instruments was performed before each test by measuring the response when the input and receiving fibers faced each other through a neutral-density filter in a black tube.

At the end of the exercise test, pen marks were made on the subject’s skin to indicate the margins of the TR-NIRS optode holders to reposition the probes for subsequent laboratory visits. The adipose tissue thickness (ATT) and muscle thickness were measured using B-mode ultrasound (SSD-3500SV, Hitachi-Aloka Medical, Tokyo) with the subject at rest and seated in an upright position. To quantify the influence of ATT on dynamic changes in TR-NIRS signals, we used the ATT correction method of Bowen et al. (2013) [11]. With this method, the tissue O_2_ saturation (S_t_O_2_) was calculated using oxy[Hb + Mb]/total[Hb + Mb].

### 2.6. Data Analysis

The peak values for gas exchange/ventilation were detected and then averaged for 30 s at the peak. Similarly, the baseline (resting) values of gas exchange/ventilation were calculated as the mean value over the final 30 s of the rest period. The output frequency of both TR-NIRS systems was set to 1 Hz and averaged post hoc to increase the signal-to-noise ratio, providing one measurement every 5 s. The baseline of each TR-NIRS measurement was calculated as the mean value of the 30 s prior to the start of the incremental exercise.

The absolute values of gas exchange/ventilation and absolute muscle deoxy[Hb + Mb], oxy[Hb + Mb], and total[Hb + Mb] measurement were then calculated every 20 W from 20 W to the maximal exercise for each subject. The value for each variable at each 20-W increment was calculated as the last 60 s from the initial 20 W to 240 W. Blood gas and metabolic profiles were obtained three times (at 120, 200, and 240 W). Since there were individual differences in the exhausted exercise duration despite the absence of a difference between the HCP and placebo groups, we made data arrangements at up to 240 W of each blood gas sample.

### 2.7. Statistical Analysis

All data are expressed as the mean ± standard deviation (SD) and were analyzed using the statistical package IBM SPSS, PC program, ver. 25.0 (IBM, Tokyo, Japan). We compared the peak values of gas exchange, overall exercise time, and mean values of parameters at rest between the HCP and placebo groups, using a paired t-test. The significance of differences in each variable was determined by a two-way analysis of variance (ANOVA), comparing supplements (HCP and placebo) × work rates (20–240 W). A *post hoc* comparison was applied by Bonferroni test for the appropriate data sets when a significant *F*-value was obtained. Probability (*p*)-values < 0.05 were considered significant.

## 3. Results

### 3.1. At Rest

The mean values of the ventilatory, acid-base, and TR-NIRS profiles are presented in Table 1. The mean values of the ventilatory parameters displayed a significantly lower
$\dot{V}$_E_ (*p* < 0.05),
$\dot{V}$O_2_ (*p* < 0.01), and
$\dot{V}$CO_2_ (*p* < 0.05) in the HCP group compared to the placebo group, whereas no significant difference in HR or R was observed between the HCP and placebo groups.

With the blood gas status related to metabolism, the HCP group showed the following significant differences from the placebo group: lower pH (HCP: 7.356 ± 0.04 vs. placebo: 7.376 ± 0.04, *p* < 0.05) and higher PCO_2_ (52.4 ± 8.3 vs. 47.4 ± 8.2 mmHg, *p* < 0.05) and HCO_3_^−^ (29.1 ± 2.2 vs. 27.5 ± 2.6 mmol∙L^−1^, *p* < 0.05). The metabolic profile as Lac or Glu was within the standard range at rest with no significant difference between the HCP and placebo groups.

The TR-NIRS profiles at the RF muscle revealed significantly lower deoxy[Hb + Mb] (*p* < 0.05) and S_t_O_2_ (*p* < 0.05) values, whereas the VL muscle showed no significant difference in either the HCP or placebo group.

### 3.2. During Incremental Exercise

All peak values of gas exchange parameters, workload, and the exhausted time were similar in the two supplement groups (Table 2). Note that the
$\dot{V}$_E_,
$\dot{V}$O_2_,
$\dot{V}$CO_2_, HR, and R responses increased in proportion to the increasing work rate (time effect, all *p* < 0.001, η^2^ = 0.917–0.993) and only
$\dot{V}$_E_ showed a significant difference between the HCP and placebo groups at some work rates (interaction effect: F(11,110) = 2.206, *p* = 0.019, η^2^ = 0.181), as illustrated in Figure 1. The metabolic parameters of
$\dot{V}$O_2_ and
$\dot{V}$CO_2_ HR and R were quite similar in the two conditions at all work rates (Figure 1B–E). P_ET_CO_2_ first increased from 20 to 120 W then decreased from 160 to 240 W despite the lack of a significant difference between the HCP and placebo groups (Figure 1F).

A main effect of supplement was noted in the form of a lower pH (supplement effect: F(1,15) = 4.879, *p* = 0.043, η^2^ = 0.245) and a higher HCO_3_^−^ (supplement effect: F(1,15) = 5.762, *p* = 0.030, η^2^ = 0.278) between the HCP and placebo groups during incremental exercise (Figure 2A,C). The change in Lac was similar between the two groups, which might be due to a greater HCO_3_^−^ value as an index of buffering capacity, which was supported by our observation of a lower AGap in the HCP group compared to the placebo group (Figure 2D).

In the VL muscles of the subjects, even though the deoxy[Hb + Mb] and total[Hb + Mb] increased with the work rates (*p* < 0.001, Figure 3A,B), the S_t_O_2_ was kept constant throughout the incremental exercise at a given work rate (time effect: F(11, 121) = 0.791, *p* = 0.648, η^2^ = 0.067, Figure 3C). The deoxy[Hb + Mb] and total[Hb + Mb] values did not differ between the HCP and placebo groups at a given work rate. In contrast, the deoxy[Hb + Mb] value in the RF muscles of the subjects was significantly greater from 200 to 240 W in the HCP group compared to the placebo group (interaction effect: F(11, 121) = 2.726, *p* = 0.004, η^2^ = 0.199, Figure 3D). The total[Hb + Mb] profile in the VL muscle was affected by deoxy[Hb + Mb], although there was not a significant interaction between the HCP and placebo groups (interaction effect: F(11, 121) = 1.781, *p* = 0.064, η^2^ = 0.13, Figure 3E). The S_t_O_2_ tended to decrease with the work rates from approx. 57% to 45% (time effect: F(11, 121) = 1.469, *p* = 0.152, η^2^ = 0.118, Figure 3F) with lower mean values in the HCP group compared to the placebo group (supplement effect: F(1, 11) = 4.405, *p* = 0.060, η^2^ = 0.286). Regarding the RF muscle, muscle deoxygenation would be promoted by the HCP supplement, providing greater deoxy[Hb + Mb] and lower muscle S_t_O_2_ values.

The slope of the relationship between the
$\dot{V}$_E_ and deoxy[Hb + Mb] profiles was steeper in the HCP group (slope = 0.367) compared to the placebo group (slope = 0.227) (Figure 4), which means that the depressed
$\dot{V}$_E_ using HCP promotes working muscle deoxygenation (especially in the RF muscle) during incremental exercise as well as rest.

## 4. Discussion

To the best of our knowledge, the present study is the first investigation to use a form such as HCP capsules as a method of administering H_2_ (instead of H_2_ water or H_2_ gas) and to comprehensively investigate the effects of HCP by examining a wide spectrum of acid-status and metabolic status values in addition to the working muscle deoxygenation measured by TR-NIRS. The main finding of this study was that H_2_ in the form of HCP after three days of intake caused a slightly but significantly lower pH and greater PCO_2_ due to hypoventilation, which might be due to the reduced ventilatory responsiveness to pH by the consumption of HCP (Figure 5). The reduction in oxidative stress with antioxidant treatment may suppress the peripheral chemoreceptor response (i.e., inhibition) more than the central chemoreceptor response (i.e., stimulation) [32]. No similar observation was made thus far in previous research examining H_2_. In fact, quite the opposite effect was observed, as H_2_-infused water caused metabolic alkalosis, decreased the blood lactate level, and lowered the rate of perceived exertion (RPE) [5,7,33].

The existing hypoventilation can be clarified by this study’s results, as the mean values of V_E_ at rest were significantly lower with HCP concomitant with a greater PCO_2_ compared to the placebo, suggesting hypoventilation, as seen by the lower
$\dot{V}$O_2_ and
$\dot{V}$CO_2_ (Table 1). In addition, a significant difference was detected between the HCP and placebo groups, showing the presence of mitigated ventilation at rest that was apparent in the blood results of lower pH, PO_2_, SO_2_, Cl, and higher PCO_2_ and HCO_3_^−^ with HCP compared to the placebo (Table 1). The decreased pH was most likely caused by the primary elevation in PCO_2_ as a result of suppressed ventilation [34] and, consequently, as lower pH caused the renal retention of bicarbonate. This is a compensatory mechanism to induce acidemia to maintain the balance pH, and the end result is an increased concentration of bicarbonate and a decreased chloride concentration [35,36].

During incremental exercise, a significantly lower
$\dot{V}$_E_ at some work rates was reflected by the ventilatory analysis values (Figure 1A) demonstrating the ongoing impact of HCP on the
$\dot{V}$_E_ response during incremental exercise as well. Despite that, the negative effect of HCP is eliminated in
$\dot{V}$CO_2_ and
$\dot{V}$O_2,_ as shown by the lack of significant changes in these parameters during exercise (Figure 1B,C); thus, the metabolic parameters increased proportionally to the work rate as a consequence of the increased respiratory and metabolic demands of the exercise [37]. At peak performance, HCP also had no effect on the exhausted exercise time or peak gas exchange parameters or on the subjects’ HR (Table 2). Therefore, it has not yet been demonstrated that the ingestion of molecular hydrogen is effective for peak performance in trained subjects.

As a result of the increased
$\dot{V}$_E_ during the progressive exercise, the Cl depletion ceased (Figure 2E) and the blood Cl was normalized [35]. However, the changes in blood pH and HCO_3_^−^ continued (Figure 2A,C), due to the continued hypoventilation despite the absence of significant differences in P_ET_CO_2_ and lactate between the HCP and placebo groups. Consequently, the decreased AGap in the HCP subjects (Figure 2D) was influenced as a result of the corrected Cl but not HCO_3_^−^. In addition, no significant difference in Lac was observed at rest or during exercise between the HCP and placebo groups, which might have been due to the higher HCO_3_^−^ acting as a buffer. Therefore, judging from the acid-base status (Figure 2A–C), it is likely that HCP enhances the lactic acid buffering capacity despite hypoventilation.

In the relationship between ROS and ventilation, we calculated the ventilatory responsiveness to pH using the mean values of
$\dot{V}$_E_ and pH, and the ventilatory responsiveness to pH was reduced by the intake of HCP compared to placebo (Figure 5). Lee et al. (2009) showed that ventilatory sensitivity to hypercapnia following antioxidant administration was significantly reduced compared to placebo in patients with sleep apnea, which we speculated was a measure of the patients’ central and peripheral chemoreflex sensitivity [38]. The reduced ventilatory responsiveness to pH or CO_2_ is associated with an increase in the CO_2_ reserve in the presence of changes in hypercapnic hypoventilation [39]. 

The LTF was reduced in patients with sleep apnea who were given an antioxidant cocktail, and it was suggested that the release of ROS may have a role in the induction and maintenance of LTF [38,40].

Wilkerson et al. (2008) also demonstrated that an intravenous administration of 10 mg of a superoxide anion scavenger to anesthetized, vagotomized, and ventilated rats before their exposure to acute hypoxia abolished the LTF of phrenic nervous activity [18], which led to lower ventilation. If LTF were active as a result of HCP, continued hyperventilation would be predicted, which is associated with ROS production [16,41]. In fact, we observed that the HCP supplementation clearly instigated lower PO_2_ and greater PCO_2_ values due to hypoventilation in our subjects, and we thus speculated that these physiological phenomena are probably caused by hypoventilation due to an excessive suppression of ROS production by HCP. The neuromodulation of LTF in humans is not well understood, and few studies have attempted to tackle this issue. However, additional research is necessary to determine the mechanisms responsible for modulating LTF in humans via antioxidant administration.

Given these findings in animals and humans, there is a possibility that the ROS would be downregulated by the administration of molecular hydrogen and cause hypoventilation and blood hypercapnia. We thus hypothesized that an excessive administration of an antioxidant of molecular hydrogen would reduce the magnitude of the ventilatory response that was associated with the increase in the CO_2_ reserve before and during exhausted exercise in humans.

With the use of TR-NIRS during exercise, we observed different reactions in the HCP and placebo subjects depending on the muscle site, which can be attributed to the differing recruitment of fast-twitch muscle fibers (which are higher in the RF muscle) and slow-twitch muscle fibers (which are higher in the VL) [10,14,42].

In the RF muscle, we observed significantly greater deoxy[Hb + Mb] at rest and higher work rates during the incremental exercise by the administration of HCP. The total[Hb + Mb] was also higher, mostly reflecting the significantly increased deoxy[Hb + Mb] between the HCP and placebo groups, with an average total increase of 12 µM. The greater deoxy[Hb + Mb] at the same work rates suggested that impairments in the
$\dot{Q}$O_2_/$\dot{V}$O_2_ ratio necessitated higher fractional O_2_ extractions to support any given change in the external work rate in this muscle. These observations suggest that the primary mechanism by which the RF achieved greater fractional O_2_ extraction at higher work rates in the HCP group was via elevated diffusive O_2_ conductance consequent to an increased microvascular [hematocrit] [14,43]. Indeed, the velocity of capillary red blood cells increases more with contractions in less-oxidative rat muscles [44], and faster red blood cell velocity is associated with a higher capillary hematocrit [45]. The RF muscle is more dependent on O_2_ extraction for a given degree of muscle activation compared to the VL muscle [43,46].

As shown in Figure 4, the greater deoxy[Hb + Mb] concentrations as a function of ventilation in the HCP subjects might be attributed to increased oxygen extraction at hypoventilation that accompanies the greater recruitment of fast-twitch fibers [47]. In addition, the S_t_O_2_ value in the RF muscle was lower with HCP compared to the placebo as a result of hypoventilation. Collectively, therefore, the present findings suggest that HCP exerted its greatest effects on the RF muscle at higher work rates where type II fiber recruitment would be expected to dominate [48].

In the VL muscle, the total[Hb + Mb] and deoxy[Hb + Mb] concentrations did not differ significantly between the placebo and HCP groups at any work rates. The S_t_O_2_ value also did not differ significantly between the two supplement groups, and this can be attributed to the relative greater presence of slow-twitch fibers in VL muscle. Slow-twitch fibers have a greater number of capillaries around each fiber and show better vasodilatory dynamic control and better oxygen extraction compared to fast-twitch fibers [42,46]. In particular, the VL muscle appears to possess a greater
$\dot{Q}$O_2_/$\dot{V}$O_2_ ratio during exercise compared to the RF muscle [14,49,50,51]. It has been suggested that differences between the VL and RF muscles may emanate, in part, from higher blood flow [52,53,54] and a greater proportion of more highly oxidative type I fibers in the VL muscle [55].

## 5. Study Limitations

In our study, the HCP supplementation provided approx. 2.5 µg/day of H_2_. However, although HCP has a smaller amount of H_2_ compared to H_2_-rich water [5,33], the presence of a compensatory respiratory response suggests an abnormal intake/absorption of H_2_, thus indicating that ingesting H_2_ in a form like HCP capsules provides a possibly better delivery as it releases H_2_ continuously in the intestinal lumen, which might make it an appealing method of administering H_2_. However, further research is warranted to test these concepts.

Some study limitations must be considered when interpreting the present results, including the small number of available investigations of hydrogen during exercise. As mentioned earlier, the present study is the first to use a form like HCP as a method of administering H_2_, and further research is needed to better comprehend the working conditions and limitations of delivery methods such as HCP. A further examination of the respiratory gas and blood sampling at rest with different durations of supplementation prior to exercise and post-exercise recovery might have shed further light on the mechanisms and actions of H_2_ in the form of HCP. Finally, we did not measure the level of ROS in the HCP intake. However, these limitations do not negate the important findings of this research.

## 6. Conclusions

H_2_-rich calcium powder supplementation, which increased the potential for antioxidant-dependent slightly lower pH at rest, resulted in significantly lower
$\dot{V}$_E_ and pH status during the incremental exercise compared to placebo. The gas exchange status of
$\dot{V}$CO_2_ and
$\dot{V}$O_2_ were not affected by HCP. In addition, the HPC induced a significantly lower O_2_ delivery/utilization ratio at the RF muscle site but not in the VL muscle, which may be explained by these sites possessing inherently different vascular and metabolic control properties, perhaps related to their fiber-type composition.

## Figures and Tables

**Figure 1 nutrients-13-00459-f001:**
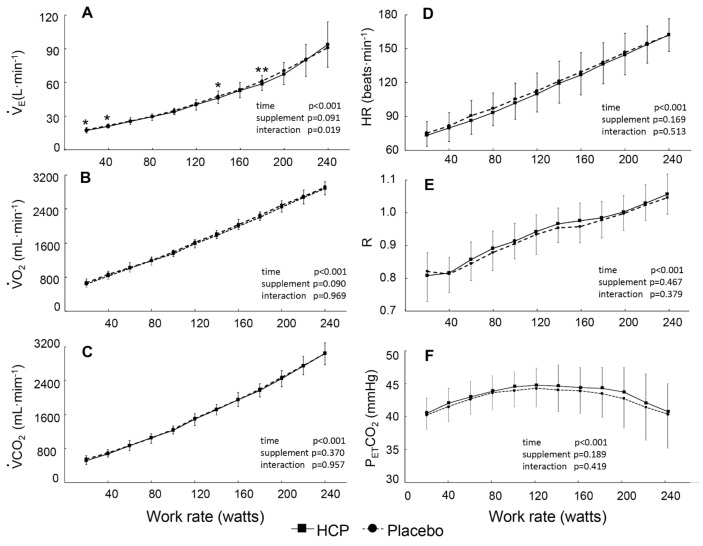
Group mean values of pulmonary ventilation (
$\dot{V}$_E_) (**A**), O_2_ uptake ($\dot{V}$O_2_) (**B**), CO_2_ output (
$\dot{V}$CO_2_) (**C**), heart rate (HR) (**D**), gas exchange ratio (R) (**E**), and end-tidal CO_2_ pressure (P_ET_CO_2_) (**F**) versus the power output during the cycle exercise in the HCP and placebo groups. Error bars: SD. The significant difference between HCP and placebo at higher work rates is shown; * *p* < 0.05, ** *p* < 0.01.

**Figure 2 nutrients-13-00459-f002:**
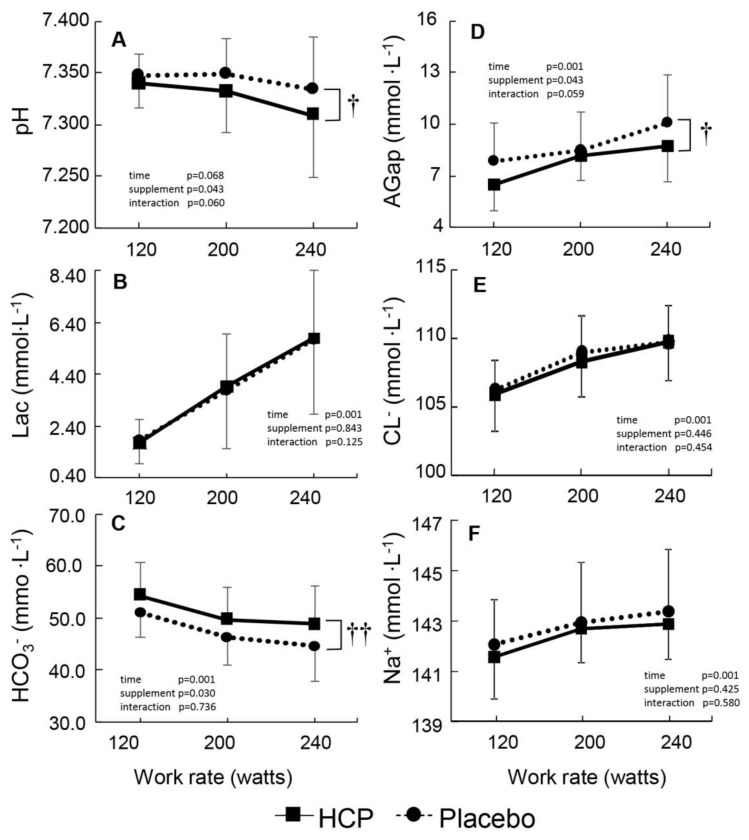
Group mean values of peripheral venous blood pH (**A**), lactate (Lac) (**B**), bicarbonate (HCO_3_^−^) (**C**), aniongap (AGap) (**D**), chloride (Cl^−^) (**E**), and sodium (Na^+^) (**F**) with the function of different work rates during the cycle exercise in the HCP and placebo groups. Error bars: SD. Significant difference between HCP and placebo at higher work rates † *p* < 0.05, †† *p* < 0.01.

**Figure 3 nutrients-13-00459-f003:**
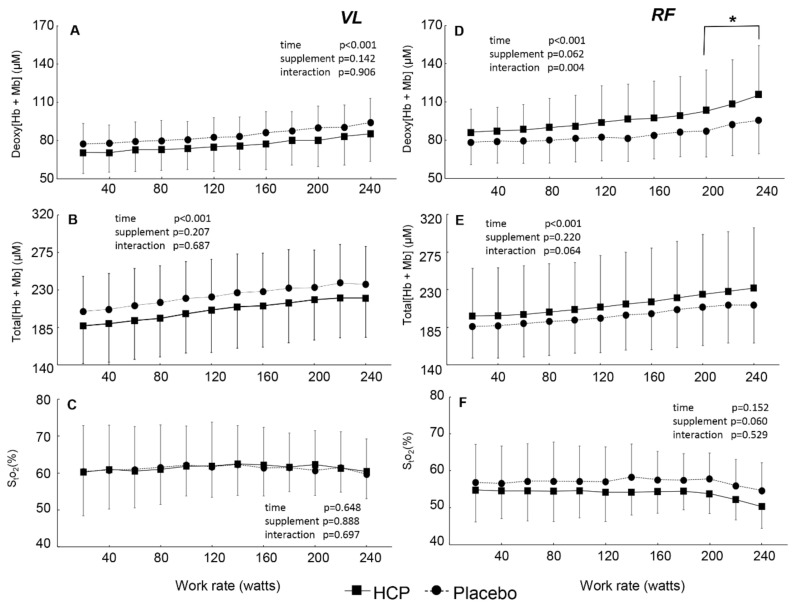
Group mean values for the vastus lateralis (VL) muscle deoxygenated hemoglobin and myoglobin concentration (deoxy[Hb + Mb]) (**A**), total hemoglobin and myoglobin concentration (total[Hb + Mb]) (**B**), tissue O_2_ saturation (S_t_O_2_) (**C**), and rectus femoris (RF) muscle deoxy[Hb + Mb] (**D**), total[Hb + Mb] (**E**), and S_t_O_2_ (**F**) versus the power output during cycle exercise in the HCP and placebo groups. Error bars: SD. Significant difference between HCP and placebo at higher work rates; * *p* < 0.05.

**Figure 4 nutrients-13-00459-f004:**
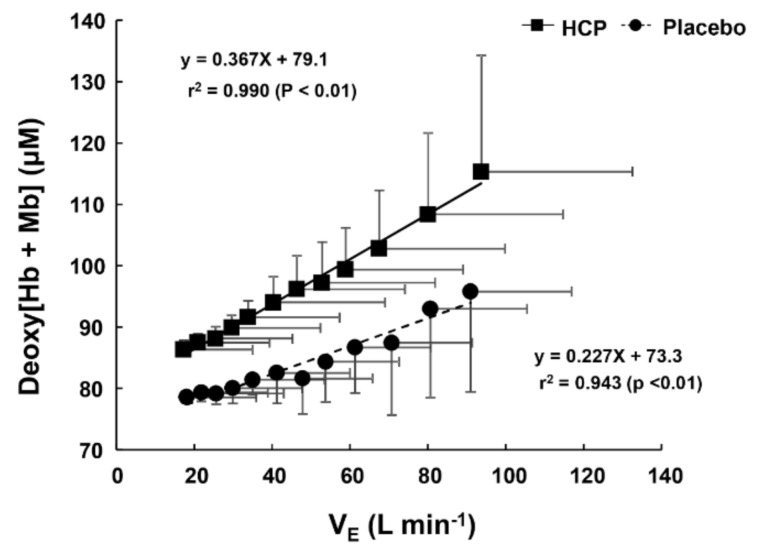
The relationship between ventilation ($\dot{V}$_E_) and the rectus femoris (RF) muscle deoxygenated hemoglobin and myoglobin concentration (deoxy[Hb + Mb]) profiles during cycle exercise in the HCP and placebo groups. Note that the regression line for
$\dot{V}$_E_–deoxy[Hb + Mb] profiles’ relationship was steeper in the HCP group (slope = 0.367) than in the placebo group (slope = 0.227). Error bars: SD.

**Figure 5 nutrients-13-00459-f005:**
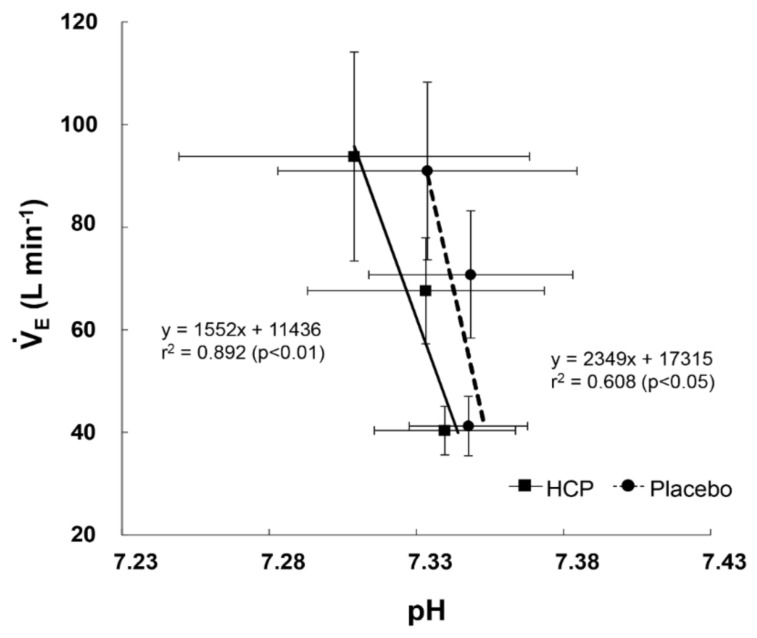
The relationship between ventilation (
$\dot{V}$_E_) and pH profiles during the cycle exercise between the HCP and placebo group. Note that the regression line for the $\dot{V}$_E_–pH profiles’ relationship was steeper in the HCP group (slope = 0.892) than in the placebo group (slope = 0.608) as a function of the reduced chemoreflex drive to pH by the administration of HCP. Error bars: SD.

**Table 1 nutrients-13-00459-t001:** Mean values of ventilatory, acid-base, and NIRS profiles at rest condition between the HCP and placebo groups.

	HCP			Placebo			*p* Value
**metabolic gas exchange**							
V_E_ (L·min^−1^)	11.8	±	3.1	13.2	±	3.2	0.02 *
VO_2_ (mL·min^−1^)	355	±	109	429	±	136	0.01 *
VCO_2_ (mL·min^−1^)	306	±	96	364	±	130	0.03 *
HR (beats·min^−1^)	75.3	±	12.3	78.5	±	15.2	0.14
R	0.9	±	0.1	0.8	±	0.1	0.22
**blood gas**							
pH	7.356	±	0.04	7.376	±	0.04	0.048 *
PO_2_ (mmHg)	43.9	±	19.3	51.0	±	17.8	0.107
PCO_2_ (mmHg)	52.4	±	8.3	47.4	±	8.2	0.026 *
HCO_3_^−^ (mmol·L^−1^)	29.1	±	2.2	27.5	±	2.6	0.041 *
SO_2_ (%)	66.8	±	25.2	76.9	±	19.7	0.051
BE(ecf) (mmol·L^−1^)	3.6	±	1.9	2.4	±	2.3	0.071
TCO_2_ (mmol·L^−1^)	30.7	±	2.5	29	±	2.8	0.041 *
Hct (%)	46	±	2.4	46	±	3.1	0.328
Hgb (g/dL)	15.8	±	0.8	15.6	±	1.1	0.313
**electrolytes**							
Na^+^ (mmol·L^−1^)	141	±	1.6	141	±	1.7	0.069
K^+^ (mmol·L^−1^)	3.8	±	0.3	4	±	0.2	0.062
Ca^2+^ (mmol·L^−1^)	1.26	±	0	1.25	±	0.0	0.471
Cl^−^ (mmol·L^−1^)	105	±	1.7	106	±	1.8	0.011 *
AGap (mmol·L^−1^)	7	±	1.6	7	±	2.2	0.444
AGapK (mmol·L^−1^)	11	±	1.6	11	±	2.3	0.365
**metabolic status**							
Lac (mmol·L^−1^)	1.13	±	0.4	1.28	±	0.6	0.312
Glu (mg·dL^−1^)	98	±	14.1	104	±	21.2	0.165
Crea (mg·dL^−1^)	0.96	±	0.1	0.92	±	0.2	0.212
**TR-NIRS in the RF muscle**							
Total[Hb + Mb] (µM)	206	±	48	201	±	37	0.469
Deoxy[Hb + Mb] (µM)	96	±	22	85	±	23	0.045 *
StO_2_ (%)	53	±	8	57	±	12	0.028 *
**TR-NIRS in the VL muscle**							
Total[Hb + Mb] (µM)	200	±	37	212	±	37	0.164
Deoxy[Hb + Mb] (µM)	76	±	20	76	±	22	0.877
StO_2_ (%)	61	±	12	63	±	12	0.222

Data are shown as mean ± standard deviation (SD). Significant difference between HCP and placebo (* *p* < 0.05).

**Table 2 nutrients-13-00459-t002:** Peak values of gas exchanges parameters, workload, and the exhausted time during incremental exercise between the HCP and placebo groups.

Metabolic Gas Exchange	HCP	Placebo	*p* Value
V_E_ (L·min^−1^)	115.2 ± 24.3	114.7 ± 28.8	0.918
VO_2_ (mL·min^−1^)	3119 ± 423	3141 ± 546	0.716
VCO_2_ (mL·min^−1^)	3496 ±576	3528 ± 702	0.721
HR (beats·min^−1^)	178.1 ± 6.9	179.2 ± 8.3	0.383
R	1.1 ± 0.1	1.1 ± 0.1	0.859
Workload (W)	270 ± 34	272 ± 29	0.430
Exhausted Time (min)	26.8 ± 4.0	26.9 ± 3.9	0.701

Data are shown as mean ± standard deviation (SD).

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
