# Peer review of "Application of Molecular Hydrogen as an Antioxidant in Responses to Ventilatory and Ergogenic Adjustments during Incremental Exercise in Humans"

_nutrients, 2021, doi:10.3390/nu13020459_

Round 1
Reviewer 1 Report
The authors conducted a novel study investigating the effects of molecular hydrogen on acid-base balance and incremental exercise. The study is well-designed and executed and manuscript is well-written. However, there are a few points that may be addressed for in the manuscript.
Subjects: The abstract states subjects were trained subjects, however no mentioning of training status is mentioned in the Subjects section. Training characteristics such as mode (aerobic [running, cycling, rowing etc.], anaerobic [HIIT, etc.], or resistance/strength, frequency (days per week), duration (minutes, hours per day), history (how many months, years of consistent training), and intensity should be described.
Experimental Overview: The breakfast before testing was standardized with the Calorie Mate blocks and barley tea or did subjects eat breakfast at home (instructed to replicate from their dietary intake report) and consumed the Calorie Mate blocks and barley tea upon arrival to the lab? It is unclear if the Calorie Mate blocks were subjects’ breakfast.
Line 339 – 340: “Peak performance in athletes” is stated, however the current study subjects were trained individuals (stated in the abstract), not athletes. Results from this study would not answer if “molecular hydrogen is effective for peak performance in athletes”. “Peak performance in athletes” can be referenced, however it should not be presented as if positive results in this study would answer that question.
Line 376: “Downregulates” likely is meant to be “downregulated”.
Discussion: There were no performance effects observed using the incremental cycling exercise to exhaustion. Do the authors have any speculations about effects of HCP in different types of exercise such as HIIT, time trial, sprint, or other anaerobic performance? HCP may provide an ergogenic effect in other types of performance. This would line up with the point of HCP exerting greater effects at higher work rates where type II fiber recruitment would dominate. This may be a point worth discussing.
Author Response
Response to reviewer 1
The authors conducted a novel study investigating the effects of molecular hydrogen on acid-base balance and incremental exercise. The study is well-designed and executed and manuscript is well-written. However, there are a few points that may be addressed for in the manuscript.
Overall response: Thank you for the positive and constructive comments about our manuscript. Addressing these comments has substantially improved the manuscript. We now present a revised version based on the comments that we hope satisfies the concerns of each of the Reviewers.
Subjects: The abstract states subjects were trained subjects, however no mentioning of training status is mentioned in the Subjects section. Training characteristics such as mode (aerobic [running, cycling, rowing etc.], anaerobic [HIIT, etc.], or resistance/strength, frequency (days per week), duration (minutes, hours per day), history (how many months, years of consistent training), and intensity should be described.
Response: We use the term "trained subject" to indicate an active member of a university sport clubs (mainly triathlons, track and field), participating mostly in aerobic sports and training regimens. Details of the training statuses were not collected, as the subjects ranged widely in training status, frequency, duration, intensity and history.
Experimental Overview: The breakfast before testing was standardized with the Calorie Mate blocks and barley tea or did subjects eat breakfast at home (instructed to replicate from their dietary intake report) and consumed the Calorie Mate blocks and barley tea upon arrival to the lab? It is unclear if the Calorie Mate blocks were subjects’ breakfast.
Response: We have added the following text beginning at line 113.
" The subjects consumed breakfast at home ≥6 h prior to the start of the experiment (and were instructed to replicate it from their dietary intake report). At 3 h before the experiment, the subjects ate a small meal consisting of the Calorie Mate (four blocks, Otsuka Pharmaceutical, Tokyo) and one bottle of caffeine-free barely tea (Healthy Mineral Barley Tea, 600 ml, ITO EN, Tokyo) that was standardized for all subjects, in order to avoid hunger and minimize fluctuations in significant blood metabolic parameters (specifically blood glucose) between/within subjects."
Line 339 – 340: “Peak performance in athletes” is stated, however the current study subjects were trained individuals (stated in the abstract), not athletes. Results from this study would not answer if “molecular hydrogen is effective for peak performance in athletes”. “Peak performance in athletes” can be referenced, however it should not be presented as if positive results in this study would answer that question.
Response: We agree and in accord with your comment, we replaced the description "peak performance in athletes" with "trained subjects". The trained subjects in this study are university athletes, not top athletes from around the world.
Line 376: “Downregulates” likely is meant to be “downregulated”.
Response: We apologize for this typo; we have corrected it.
Discussion: There were no performance effects observed using the incremental cycling exercise to exhaustion. Do the authors have any speculations about effects of HCP in different types of exercise such as HIIT, time trial, sprint, or other anaerobic performance? HCP may provide an ergogenic effect in other types of performance. This would line up with the point of HCP exerting greater effects at higher work rates where type II fiber recruitment would dominate. This may be a point worth discussing.
Response: Thank you very much for this valuable comment. As we briefly mentioned in the text about study limitations, further research is needed to better comprehend the working conditions and limitations of delivery methods such as HCP. Recently, we investigated that the HCP supplement improves the average power output during anaerobic performance with a Wingate test protocol, just like HIT and time trials (unpubl. data). The HCP induced greater performance at higher work rates, where the recruitment of fast-twitch muscle fibers is dominant. We thus suspect that HCP might have an ergogenic effect in anaerobic performance testing, but further examination is needed to test this conjecture.

Reviewer 2 Report
Title: Application of molecular hydrogen as an antioxidant in responses to ventilatory and ergogenic adjustments during incremental exercise in humans.
Decision: The authors have tested the ventilatory and ergogenic changes during incremental exercise in humans in response application of molecular hydrogen. The authors have done an interesting study, but I have some concerns about the interpretation of the results of this study.
Below are some major and minor comments the authors may wish to consider.
Major Comments.
- This study should start by showing the level of H2 after the application of the H2 capsule. The authors showed that H2 capsule decreased pH level like inducing acidosis more than placebo during exercise. And it is different from the results of the previous study, so it is possible H2 capsule is differently working with H2 water via an unknown mechanism. Authors just assume that the level is higher than the application of H2 water or others, but the evidence is necessary.
- As you know oxidative fiber dominantly using oxygen to make ATP, and production of ROS is fate during oxidative phosphorylation. The authors concluded that H2 application decreased oxygen delivery and utilization, which is the cause of the reduction in ROS.Therefore, authors should re-interpret results.
- Authors interpreted many part that H2 is antioxidant, but I don’t think H2 capsule is an antioxidant according to the results in this study, and I can’t find any evidence that H2 capsule is antioxidant. Previous studies have shown that H2 is an antioxidant, but the results in this manuscript were different from the previous study. Thus, the author should reconsider the interpretation on H2 capsules as antioxidants.
Minor comments
- Line 55, “buffering lactate production”, the meaning of buffering is a little ambiguous, H2 decreases lactate production? (H2 decreases glucose metabolism?) Or H2 promotes the conversion of lactic acid to lactate?
- Line 70-80, I think the logic of the hypothesis that of how H2 can reduce ROS is lacking. Again, lower consumption of oxygen would downregulate ROS production, it is not antioxidant effects. There is required additional explanation.
- Line 142, “Aniongap (AGap) and Aniongap (AGapK)”, same words and different definitions.
- 1A, is this the best way to show the difference of VE between HCP and Placebo? It hard to see the difference between both groups.
- Line 307-310, CO2 sensitivity of what? And for what? Please complete the sentence.
- Line 346-349, Lower oxygen delivery could induce the higher production of lactate, so higher HCO3- by H2 capsule could well buffer lactic acid, but I think higher HCO3- could not compensate for lower pH during exercise according to Fig. 2A, pH still lower during exercise.
- Line 376, it is a typo “would be downregulates”.
Author Response
Response to reviewer 2
Decision: The authors have tested the ventilatory and ergogenic changes during incremental exercise in humans in response application of molecular hydrogen. The authors have done an interesting study, but I have some concerns about the interpretation of the results of this study.
Overall response: Thank you for the critical and constructive comments about our manuscript. Addressing these comments has substantially improved the manuscript. We now present a revised version based on the comments that we hope satisfies the concerns of each of the Reviewers.
Below are some major and minor comments the authors may wish to consider.
Major Comments.
- This study should start by showing the level of H2 after the application of the H2 capsule. The authors showed that H2 capsule decreased pH level like inducing acidosis more than placebo during exercise. And it is different from the results of the previous study, so it is possible H2 capsule is differently working with H2 water via an unknown mechanism. Authors just assume that the level is higher than the application of H2 water or others, but the evidence is necessary.
Response: We had written that "HCP has a smaller amount of H2 compared to H2-rich water [5,33]" at line 418, and thus our hypothesis is consistent with yours. We realized the necessity of evidence here, but because it is very difficult to measure the H2 gas release/absorption amount in the body for both H2 water and HCP and since there is no reference for this account, we cannot provide this evidence at the present time.
It is possible that the HCP/H2 capsule functions based on a different mechanism than that of H2 water in the pathways affecting acid-base regulation. However, as both media are acting only as carriers to deliver H2 gas to the body by diffusion of the gas through the intestinal tract without affecting the molecular structure of H2, it is reasonable to speculate that the different reactions to H2 water and the HCP/H2 capsule is due to the variation in the amount of H2 delivered. In addition, we think it is possible that a single dose of H2 may be different from the 3-day continuous intake used in this study. There are currently very few studies examining H2 water. Since the present study is the first examination of the HCP/H2 capsule, this result cannot be confirmed without additional research.
- As you know oxidative fiber dominantly using oxygen to make ATP, and production of ROS is fate during oxidative phosphorylation. The authors concluded that H2 application decreased oxygen delivery and utilization, which is the cause of the reduction in ROS. Therefore, authors should re-interpret results.
Response: Our apologies; the combined respiration and muscle metabolism complicated the scenario and made it difficult to understand the working muscle oxygenation and reduction in ROS. HCP-induced hypoventilation would lead to lower pH, and then the lower pH would induce a impaired balance between O2 delivery and O2 utilization in working muscle (i.e., RF muscle). Consequently, the regression line for the VE-deoxy[Hb+Mb] profile was steeper in the HCP group (Fig 4). At the same time, the sensitivity of peripheral chemoreceptors might be suppressed by HCP (Fig.5). Thus, the H2 application did not directly decrease oxygen delivery and utilization; the application of H2 would lead to hypoventilation and lower pH, and then impairs the balance between O2 delivery and O2 utilization (i.e., the O2 delivery/utilization ratio) in working muscle.
- Authors interpreted many part that H2 is antioxidant, but I don’t think H2 capsule is an antioxidant according to the results in this study, and I can’t find any evidence that H2 capsule is antioxidant. Previous studies have shown that H2 is an antioxidant, but the results in this manuscript were different from the previous study. Thus, the author should reconsider the interpretation on H2 capsules as antioxidants.
Response: Previous studies have shown that H2 has an antioxidant effect when used in either a gas form or the H2 water form (4,6) and we thus feel that it is reasonable to speculate that H2 gas from HCP will also demonstrate the same antioxidant effects. As explained in our manuscript, we compared the difference between our findings and the previous research in terms of acid-base regulation, not antioxidant capacity. The earlier cited studies that obtained different results did not examine ROS or the antioxidant effect of H2, and those studies both focused on physiological effects in term of exercise performance.
Minor comments
- Line 55, “buffering lactate production”, the meaning of buffering is a little ambiguous, H2 decreases lactate production? (H2 decreases glucose metabolism?) Or H2 promotes the conversion of lactic acid to lactate?
Response: We have replaced “lactate production” with “lactic acid”.
- Line 70-80, I think the logic of the hypothesis that of how H2 can reduce ROS is lacking. Again, lower consumption of oxygen would downregulate ROS production, it is not antioxidant effects. There is required additional explanation.
Response: As we noted above, other studies demonstrated that H2 has an antioxidant effect when used in either a gas form or the H2 water form (4,6) and we thus feel that it's reasonable to hypothesize that H2 gas from HCP will also demonstrate the same antioxidant effects. In respiratory physiology, antioxidant intervention would mitigate the ventilation, resulting in hypoventilation and blood hypercapnia (16,17,18). Many studies using animal experiments have confirmed that ROS is the main factor in the ability of antioxidant administration to cause hypoventilation. Our study's hypothesis thus involves a complex interplay between clinical respiration and energy metabolism.
- Line 142, “Aniongap (AGap) and Aniongap (AGapK)”, same words and different definitions.
Response: We have replaced the term "Aniongap (AGapK)" with "Aniongap, potassium (AGapK)" at L 142.
- 1A, is this the best way to show the difference of VE between HCP and Placebo? It hard to see the difference between both groups.
Response: We agree with your observation about the difference between HCP and placebo, but we used a two-way ANOVA and post-hoc Bonferroni tests to determine significant differences between HCP and placebo. It's possible that it may simply be difficult to observe the differences at 140 and 180 watts between HCP and placebo because of the large scale of VE on the vertical axis.
- Line 307-310, CO2 sensitivity of what? And for what? Please complete the sentence.
Response: The description "CO2 sensitivity" was inadequate wording and has been deleted.
- Line 346-349, Lower oxygen delivery could induce the higher production of lactate, so higher HCO3- by H2 capsule could well buffer lactic acid, but I think higher HCO3- could not compensate for lower pH during exercise according to Fig. 2A, pH still lower during exercise.
Response: In accord with your suggestion, the wording "and compensating for lower pH" was deleted.
- Line 376, it is a typo “would be downregulates”.
Response: We have corrected this.

Reviewer 3 Report
The study by Alharbi et al studied the effect of molecular hydrogen (H2) supplementation on acid-base status, 12 pulmonary gas exchange responses, and local muscle oxygenation during incremental exercise. The authors reported that HCP supplementation decreased VE, VCO2, but increased HCO3−, PCO2. During exercise, a lower pH and VE was observed in the HCP group, together with a higher HCO3− . The study is unique, and the data is interesting.
A few comments for the authors:
- Discussion should be improved.
- Please go through the manuscript thoroughly, for errors and types, for example, Line 224 and Line 322
- Have the authors considered checking the ROS level in participants? If not, at least this should be mentioned as a limitation of the study.
- How did the authors justify the dose and timing of HCP supplementation?
- Table 2, how about the peak work load?
Author Response
Response to reviewer 3
Comments and Suggestions for Authors
The study by Alharbi et al studied the effect of molecular hydrogen (H2) supplementation on acid-base status, 12 pulmonary gas exchange responses, and local muscle oxygenation during incremental exercise. The authors reported that HCP supplementation decreased VE, VCO2, but increased HCO3−, PCO2. During exercise, a lower pH and VE was observed in the HCP group, together with a higher HCO3−.The study is unique, and the data is interesting.
Response: Thank you for the positive and constructive comments about our manuscript. Addressing these comments has substantially improved the manuscript. We now present a revised version based on the comments that we hope satisfies the concerns of each of the Reviewers.
A few comments for the authors:
- Discussion should be improved.
Response: Your comment and those of Reviewers #1 and #2 were taken into account to improve the Discussion.
  2.Please go through the manuscript thoroughly, for errors and types, for example, Line 224 and Line 322
Response: The errors you pointed out at L224 and L322 have been corrected.
  3.Have the authors considered checking the ROS level in participants? If not, at least this should be mentioned as a limitation of the study.
Response: We did not measure the level of ROS in the HCP intake. In accord with your comment, the information about the examination of ROS level has been added to the manuscript's section about study limitations.
4. How did the authors justify the dose and timing of HCP supplementation?
Response: As mentioned earlier, there were no research examining a H2 delivery form like HCP, and there are no established criteria for a minimum effective dose or a maximum safe dose for H2. We thus selected the dose and timing of HCP supplementation in this study based on the results of preliminary experiments and on the supplement manufacturer's testing data (unpubl. data).
5. Table 2, how about the peak work load?
Response: The mean values of the peak workload are shown in Table 2. Peak workload: HCP= 270 ± 34 W, Placebo: 272 ± 29 W, P=0.430.

Round 2
Reviewer 2 Report
I don't have further comments.
Reviewer 3 Report
The authors have adequately addressed my comments.